# Weakly Supervised Dense Event Captioning in Videos

**Xuguang Duan**[*1], **Wenbing Huang**[*2], **Chuang Gan**[3], **Jingdong Wang**[4],

**Wenwu Zhu**[1], **Junzhou Huang**[2]
[1] Tsinghua University, Beijing, China; [2] Tencent AI Lab. ;
[3] MIT-IBM Watson AI Lab; [4] Microsoft Research Asia, Beijing, China;
duan_xg@outlook.com, hwenbing@126.com, ganchuang1990@gmail.com,
jingdw@microsoft.com, wwzhu@tsinghua.edu.cn,joehhuang@tencent.com

## Abstract

Dense event captioning aims to detect and describe all events of interest contained in a video. Despite the advanced development in this area, existing methods tackle this task by making use of dense temporal annotations, which is dramatically source-consuming. This paper formulates a new problem: weakly supervised dense event captioning, which does not require temporal segment annotations for model training. Our solution is based on the one-to-one correspondence assumption, each caption describes one temporal segment, and each temporal segment has one caption, which holds in current benchmark datasets and most real-world cases. We decompose the problem into a pair of dual problems: event captioning and sentence localization and present a cycle system to train our model. Extensive experimental results are provided to demonstrate the ability of our model on both dense event captioning and sentence localization in videos.

## 1 Introduction

Dramatic improvements have been made on video understanding due to the development of deep neural networks and large-scale video datasets [1, 2, 3]. Among the wide variety of applications on video understanding, the video captioning task is attracting more and more interests in recent years [4, 5, 6, 7, 8, 9, 10, 11]. In video captioning, the machine is required to describe the video content in the natural language form, which makes it more meticulous and thus challenging compared to other tasks describing the video content using a few tags or labels, such as video classification and action detection [12, 13].

The current trend on video captioning is to perform Dense Event Captioning (DEC, also called *Dense-Captioning Event in videos* in [10]). As one video usually contains more than one event of interest, the goal of DEC is to locate all events in the video and perform captioning for each of them. Clearly, such dense captioning enriches the information we obtained and is beneficial for more in-depth video analysis. Nevertheless, to achieve this goal, we need to collect the caption annotation for each event along with its temporal segment coordinate (i.e., the start and end times) for network training, which is source-consuming and impractical.

In this paper, we introduce a new problem, Weakly Supervised Dense Event Captioning (WS-DEC)[2], which aims at dense event captioning only using the caption annotations for training. In the training

---

[*]denotes equal contributions. This paper was done when Xuguang Duan was served as a research intern in Tencent AI Lab. Wenwu Zhu is the corresponding author.

[2]More specifically, the term "weakly" in our paper refers to the incompleteness of the supervision rather than the amount of information.

dataset, only a paragraph or a set of sentences is available to describe each video, but the temporal segment coordinate of each event and its correspondence to the captioning sentence is not given. For testing, the model is able to detect all events of interest and provides the caption for each event. One obvious advantage of the weak supervision is the significant reduction of the annotation cost. This benefit becomes more demanded if we attempt to make use of the videos in the wild (e.g. the videos on the web) to enlarge the training set.

We solve the problem by unitizing the one-to-one correspondence assumption: each caption describes one temporal segment, and each temporal segment has one caption. We decompose the problem into a cycle of dual problems: caption generation and sentence localization. During the training phase, we perform sentence localization from the given caption annotation, to obtain the associated segment that is then fed to the caption generator to reconstruct the caption back. The objective is to minimize the reconstruction error. Our cycle process repeatedly optimizes caption generator and sentence localizer without any ground-truth segment. During the testing phase, it is infeasible to apply the cycle process in the same way as the training phase, as the caption is unknown. Instead, we first perform caption generation on a bunch of randomly initialized candidate segments and then map the resulting captions back to the segment space. The output segments by this cycle process will get closer to the ground-truths if certain properties are satisfied. We thus formulate an extra loss for the training to enforce our model to meet these properties. Based on the detected segment, we are able to perform event captioning on it, and thus achieve the goal of dense event captioning.

We summarize our contributions as follow. **I.** We propose to solve the DEC task without the need of temporal segments annotation, thus introduce a new problem WS-DEC, aiming at making use of the huge amount of data in the web and thus reducing the cost of annotation. **II.** We develop a flexible and efficient method to address WS-DEC by exploring the one-to-one correspondence between the temporal segment and event caption. **III.** We evaluate the performance of our approach on the widely-used benchmark ActivityNet Captions [10]. The experimental results verify the effectiveness of our method regarding the dense event captioning ability and sentence localization accuracy.

## 2 Related Work

We briefly review recent advances on video captioning, dense even captioning and sentence localization in videos in the next few paragraphs.

**Video captioning.** Early researchers simply aggregate frame-level features by mean pooling and then use similar pipelines as image captioning [4] to generate caption sentences. This mean-pooling strategy works well for short video clips, but will easily crash with the increase of video length. Recurrent Neural Networks (RNNs) along with attention mechanisms are thus employed [5, 6, 7, 8], among which S2VT[7] exhibits more desirable efficiency and flexibility. Since a single sentence is far from enough to describe the dynamics of untrimmed real-world video, some researchers attempt to generate multiple sentences or a paragraph to describe the given video [9, 14, 15]. Among them, the work by [15] aims at providing diverse captions corresponded to different spatial regions in a weakly supervised manner. Despite the similar weakly-supervised setting to this work, our paper differently is to localize different events temporally and perform captioning for each detected event, which generates descriptions based on meaningful events instead of bewildering visual features.

**Dense Event Captioning.** Recent attention have been paid on dense event captioning in videos [10, 11]. Current works all follow the "detection and description" framework. The model proposed by [10] resorts to the DAP method[16] for event detection and enhance the caption ability by applying the context-aware S2VT[7]. Meanwhile, [11] employs a grouping schema based on their previous video highlight detector[17] to perform event detection, and the attribute-augmented LSTM (LSTM-A)[18] for caption generation. Most recently, [19, 20] try to boost the event proposal with generated sentence, while [21] tries to leverage bidirectional SST[22] instead of DAP[16]. Also, [21] proposes to use bidirectional attention for dense captioning. In contrast to these fully-supervised works, we address the task without the guidance of temporal segments during training. Specifically, instead of detecting all event using the one-to-many-mapping event detector[23, 22], we try to localize them one by one using our sentence localizer and caption generator.

**Sentence localization in videos.** Localizing sentence in videos is constrained to certain visual domains (*e.g.*, kitchen environment) in the early stage[24, 25, 26]. Due to the development of deep learning, several models have been proposed to work on real-world videos [27, 28, 29]. The

approaches by [27, 28] are categorized into the typical two-stage framework as "scan and localize". To elaborate a bit, the work by [27] employs a Moment Context Network(MCN) for matching candidate video clip and sentence query, while the model in [28] proposes a Cross-modal Temporal Regression Localizer (CTRL) to make use of coarsely sampled clips for computation reduction. In contrast, [29] opens up a different direction by regressing the temporal coordinate given learned video representation and sentence representation. In our framework, the sentence localization is originally formulated as an intermediate task to enable weakly supervised training for dense event captioning. Actually, our model also provides an unsupervised solution to sentence localization.

## 3 The Proposed Method

We start this section by presenting the fundamental formulation of our method and follow it up with providing the details on model architecture.

**Notations.** Prior to further introduction, we first provide the key notations used in this work. We denote the given video by $\boldsymbol{V} = (\boldsymbol{v}_1, \boldsymbol{v}_2, \cdots, \boldsymbol{v}_T)$ with $\boldsymbol{v}_t$ indexing the image frame at time $t$. We define the event of interest as a temporally-continues segment of $\boldsymbol{V}$ and denote all the events by their temporal coordinate as $\{\boldsymbol{S}_i = (m_i, w_i)\}_i^N$, where $N$ is the number of events, $m_i$ and $w_i$ denote the temporal center and width, respectively. The temporal coordinates for all events are normalized to be within $[0, 1]$ throughout this paper. Let the caption for the segment $\boldsymbol{S}_i$ be the sentence $\boldsymbol{C}_i = \{\boldsymbol{c}_{ij}\}_{j=0}^{T_c}$ where $\boldsymbol{c}_{ij}$ denotes the $j$-th word, and $T_c$ is the length of caption sentence.

### 3.1 Formulation

Formally, the conventional event captioning models [10, 11] first locate the temporal segments $\{\boldsymbol{S}_i\}_i^N$ of the events by the event proposal module, and then generate the caption $\boldsymbol{C}_i$ for each segment $\boldsymbol{S}_i$ through the caption generator. Here, for our weak supervision, the segment labels are unprovided and only the caption sentences (could be multiple for a single video) are available.

The biggest difficulty of our task lies in that it's impossible to perform weakly supervised event proposal which in nature is a one-to-many mapping problem and is too noisy for weakly learning. Instead, we try a novel new direction that makes use the bidirectional one-to-one mapping between caption sentence and temporal segment. Formally, we formulate a pair of dual tasks of *sentence localization* and *event captioning*. Conditioned on a target video $\boldsymbol{V}$, the dual tasks are defined as:

- **Sentence localization**: this task is to localize segment $\boldsymbol{S}_i$ corresponded to the given caption $\boldsymbol{C}_i$, *i.e.*, learning the mapping $l_{\theta_1} : (\boldsymbol{V}, \boldsymbol{C}_i) \rightarrow \boldsymbol{S}_i$, associated with parameter $\theta_1$;
- **Event Captioning**: the event captioning inversely generates caption $\boldsymbol{C}_i$ for the given segment $\boldsymbol{S}_i$, *i.e.*, learning the function $g_{\theta_2} : (\boldsymbol{V}, \boldsymbol{S}_i) \rightarrow \boldsymbol{C}_i$, associated with parameter $\theta_2$.

The dual problems exist simultaneously once the correspondence between $\boldsymbol{S}_i$ and $\boldsymbol{C}_i$ is one-to-one, which is the case in our problem for that $\boldsymbol{S}_i$ and $\boldsymbol{C}_i$ are tied together by their corresponding event.

If we nest these dual functions together, then any valid caption and segment pair $(\boldsymbol{C}_i, \boldsymbol{S}_i)$ becomes a fixed-point solution of the following functions:

$$\boldsymbol{C}_i = g_{\theta_2}(\boldsymbol{V}, l_{\theta_1}(\boldsymbol{V}, \boldsymbol{C}_i)), \tag{1}$$
$$\boldsymbol{S}_i = l_{\theta_1}(\boldsymbol{V}, g_{\theta_2}(\boldsymbol{V}, \boldsymbol{S}_i)). \tag{2}$$

More interestingly, Eq. (1) derives an auto-encoder for $\boldsymbol{C}_i$ where the segment $\boldsymbol{S}_i$ gets vanished. This gives us a solution to train the parameters of both functions of $l$ and $g$, by formulating the loss as

$$\mathcal{L}_c = \text{dist}(\boldsymbol{C}_i, g_{\theta_2}(\boldsymbol{V}, l_{\theta_1}(\boldsymbol{V}, \boldsymbol{C}_i))), \tag{3}$$

where $dist(\cdot, \cdot)$ is a loss distance.

A remaining issue that it is still infeasible to perform dense event captioning in the testing phase by applying $l_{\theta_1}$ or $g_{\theta_2}$ since both the temporal segment and caption sentence are unknown. To tackle the testing issue, we introduce the concept of the fixed-point iteration [30] as follow.

**Proposition 1** (Fixed-Point-Iteration)**.** *We define the iteration as*

$$\boldsymbol{S}(t + 1) = l_{\theta_1}(\boldsymbol{V}, g_{\theta_2}(\boldsymbol{V}, \boldsymbol{S}(t))), \tag{4}$$

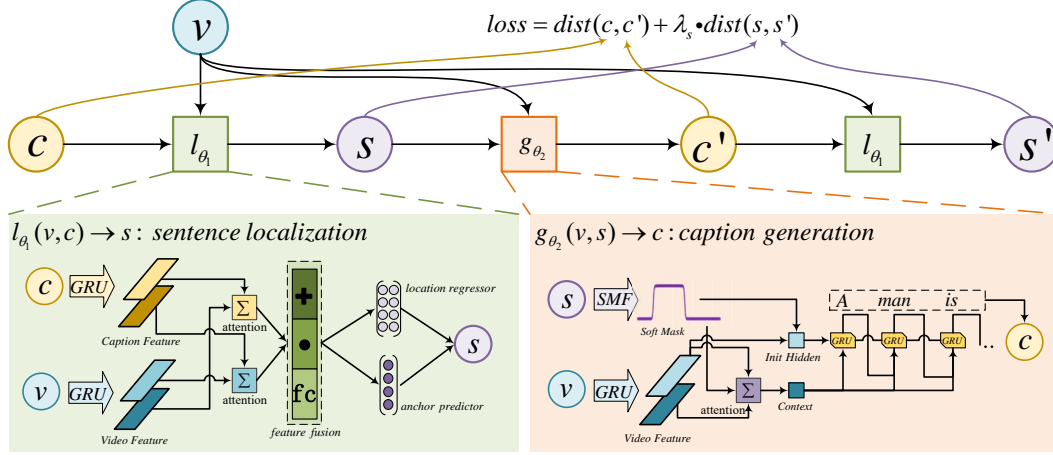

Figure 1: Model structure and training connections. Our model is composed of a sentence localizer and a caption generator. For training, the video and all event descriptions are available. We feed the video and one of its event descriptions to the sentence localizer to obtain a temporal segment prediction, and then the temporal segment is used to regenerate the caption sentence, and to relocate the temporal segment. The trained dual system is used to generate dense event caption with random temporal segments in the test phase.

where $S(t)$ will converge to the fixed-point solution i.e. $S^* = l_{\theta_1}(V, g_{\theta_2}(V, S^*))$, if there exists a sufficiently small $\epsilon > 0$ satisfying $\|S(0) - S^*\| \le \epsilon$ and the function $l_{\theta_1}(V, g_{\theta_2}(V, S))$ is locally Lipschitz continuous around $S^*$ with Lipschitz constant $L < 1$.

Note that the proof has already been derived previously. For better readability, we include them in the supplementary material.

With the application of the fixed-point-iteration, we can solve the event captioning task without any caption or segment during testing. We sample a random bunch of candidate segments $\{S_i^{(r)}\}_i^{N_r}$ for the target video as initial guesses, and then perform the iteration in Eq. (4) on these candidates. After sufficient iterations, the outputs will converge to the fixed-point solutions (*i.e.* the valid segments) $S^*$. In our experiments, we only use one-round iteration by $S_i' = l_{\theta_1}(V, g_{\theta_2}(V, S_i^{(r)}))$ and find it sufficient to deliver promising results. With the refined segments $\{S_i'\}_{i=0}^{N'}$ at hand, we are able to generate the captions as $\{C_i = g_{\theta_2}(V, S_i')\}_{i=0}^{N'}$ and thus solve the dense event captioning task.

As introduced afterward, both $l_{\theta_1}$ and $g_{\theta_2}$ are stacked by multiple neural layers which not naturally satisfy the local-Lipschitz-continuity in Proposition 1. We thus apply the idea of denoising auto-encoder in [31], where we generate noisy data by adding a Gaussian noise to the training data and minimize the reconstruction of the noisy data to the true ones. Explicitly, we enforce the temporal segments around the true data to converge to the fixed-point solutions by one-round iteration. Recalling that for weakly supervised constraint, we do not have the ground-truth segments during training, we thus apply $l_{\theta_1}(V, C_i)$ as the approximated segment, and minimize the following loss:

$$\mathcal{L}_s = dist(l_{\theta_1}(V, C_i), l_{\theta_1}(V, g_{\theta_2}(V, \varepsilon_i + l_{\theta_1}(V, C_i)))), \tag{5}$$

where $\varepsilon_i \sim \mathbb{N}(0, \sigma)$ is a Gaussian noise. The Gaussian smooth (Eq. (5)) does not theoretically hold the Lipschitz continuity, but it practically enforces the random proposals to converge to the positive segments as verified by our experiments.

By combining Eq.(3) and Eq.(5), we obtain the hybrid loss as

$$\mathcal{L} = \mathcal{L}_c + \lambda_s \mathcal{L}_s, \tag{6}$$

where $\lambda_s$ is the trade-off parameter.

## 3.2 Network Design

The core of our framework as illustrated in Fig. 1 consists of a **Sentence Localizer** (*i.e.* $l_{\theta_1}(\boldsymbol{V}, \boldsymbol{C})$) and a **Caption Generator** (*i.e.* $g_{\theta_2}(\boldsymbol{V}, \boldsymbol{S})$). Any differential model can be applied to formulate the sentence localizer and caption generator. Here, we introduce the ones that we use. Besides, we omit the RNN-based video and sentence feature extractors, leaving the details of them in the supplementary material. In the following several paragraph, suppose we have obtained the features $\boldsymbol{V} = \{\boldsymbol{v}_t \in \mathbb{R}^k\}_{t=0}^{T_v}$, hidden states $\{\boldsymbol{h}_t^{(v)} \in \mathbb{R}^k\}_{t=0}^{T_v}$ for each video, and the features $\boldsymbol{C} = \{\boldsymbol{c}_t \in \mathbb{R}^k\}_{t=0}^{T_c}$, hidden states $\{\boldsymbol{h}_t^{(c)} \in \mathbb{R}^k\}_{t=0}^{T_c}$ for each caption sentence. $T_v$ and $T_c$ are the lengths of the video and caption.

**Sentence Localizer.** Performing localization requires to model the correspondence between the video and caption. We absorb the ideas from [29, 28], and propose a cross-attention multi-model feature fusion framework. Here, we develop a novel attention mechanism named as *Crossing Attention*, which contains two sub-attention computations. The first one computes the attention between the final hidden state of the video and the caption feature at each time step, namely,

$$\boldsymbol{f}_c = \mathrm{softmax}((\boldsymbol{h}_{T_v}^{(v)})^{\mathrm{T}} \boldsymbol{A}_c \boldsymbol{C}) \boldsymbol{C}^{\mathrm{T}} \tag{7}$$

where $()^{\mathrm{T}}$ denotes the matrix transposition and $\boldsymbol{A}_c \in \mathcal{R}^{k \times k}$ is the learnable matrix. The other one is to calculate the attention between the final hidden state of the caption and the video features, *i.e.*,

$$\boldsymbol{f}_v = \mathrm{softmax}((\boldsymbol{h}_{T_c}^{(c)})^{\mathrm{T}} \boldsymbol{A}_v \boldsymbol{V}) \boldsymbol{V}^{\mathrm{T}} \tag{8}$$

where $\boldsymbol{A}_v \in \mathcal{R}^{k \times k}$ is the learnable matrix.

Then, we apply the multi-model feature fusion layer in [28] to fuse two sub-attentions as

$$\boldsymbol{f}_{cv} = (\boldsymbol{f}_c + \boldsymbol{f}_v) \| (\boldsymbol{f}_c \cdot \boldsymbol{f}_v) \| \mathrm{FC}(\boldsymbol{f}_s \| \boldsymbol{f}_v), \tag{9}$$

where $\cdot$ is the element-wise multiplication, $\mathrm{FC}(\cdot)$ is a Fully-Connected (FC) layer, and $\|$ denotes the column-wise concatenation.

One can regress the temporal segment directly by adding an FC layer on the mixed feature $\boldsymbol{f}_{cv}$, which however is easy to get suck in local minimums if the initial output is far away from the valid segment. To allow our prediction to move between two distant locations efficiently, we first relax the regression problem to a classification task. Particularly, we evenly divide the input video into multiple anchor segments under multiple scales, and train a FC layer on the $\boldsymbol{f}_{cv}$ to predict the best anchor that produces the highest Meteor score [32] of the generated caption sentence. We then conduct regression around the best anchor that gives the highest score. Formally, we attain

$$\boldsymbol{S} = \boldsymbol{S}^{(a)} + \Delta \boldsymbol{S}, \tag{10}$$

where $\boldsymbol{S}^{(a)}$ is the best anchor segment and $\Delta \boldsymbol{S} = (\Delta m, \Delta w)$ are the regression output by performing a FC layer on $\boldsymbol{f}_{cv}$.

**Caption Generator.** Given the temporal segment, we can perform captioning on the frames clipped from the video. However, such clipping operation is non-differential, making it intractable for end-to-end training. Here, we perform a soft clipping by defining a continues mask function with respect to the time $t$. This mask is defined by

$$M(t, \boldsymbol{S}) = \mathrm{Sig}(-K(t - m + w/2)) - \mathrm{Sig}(-K(t - m - w/2)), \tag{11}$$

where $\boldsymbol{S} = (m, w)$ is the temporal segment, $K$ is the scaling factor, and $\mathrm{Sig}(\cdot)$ is the sigmoid function. When $K$ is large enough, the mask function becomes a step function whose value is zero exact for the region $[m - w/2, m + w/2]$. The conventional mean-pooling feature of clipped frames are then equal to the weighted sum of the video features by the mask after normalization, *i.e.*,

$$\boldsymbol{v}' = \sum_{t=1}^{T_v} M(t, \boldsymbol{S}) \cdot \boldsymbol{v}_t / \sum_{t=1}^{T_v} M(t, \boldsymbol{S}). \tag{12}$$

Regarding $\boldsymbol{v}'$ as context, and $\boldsymbol{h}_{m+w/2}^{(v)}$ as initial hidden state, RNN is applied to generate the caption:

$$\{\bar{\boldsymbol{c}}_t\}_{t=1}^{T_c} = RNN(\boldsymbol{v}', \boldsymbol{h}_{m+w/2}^{(v)}). \tag{13}$$

**Loss Function** The loss function in Eq. (6) contains two terms, $\mathcal{L}_c$ and $\mathcal{L}_s$.

$\mathcal{L}_c$ is used to minimize the distance between the ground-truth $C = \{c_i\}_{i=0}^{T_c}$ and our prediction $\bar{C} = \{\bar{c}_i\}_{i=0}^{T_c}$. We apply cross-enctropy loss as follow(or say, perplexity loss):

$$\mathcal{L}_c = -\sum\nolimits_{t=1}^{T_c} c_t \cdot \log(\bar{c}_t | c_0 : c_{t-1}). \tag{14}$$

$\mathcal{L}_s$ is applied to compare the difference between $S = (m, w)$ and $S' = (m', w')$ as illustrated in Fig. 1, which is implemented by the $\ell_2$ norm as

$$\mathcal{L}_s = (m - m')^2 + (w - w')^2. \tag{15}$$

As metioned in Eq. (10), we further train the sentence localizer to predict the best anchor segment by adding a soft-max layer on the mixed feature $f_{cv}$ in Eq. (9). We define the one-hot label as $y = [y_1, \cdots, y_{N_a}]$ where $y_j = 1$ if the $j$-th anchor segment is the best one, otherwise $y_j = 0$. Suppose our prediction output is $p = [p_1, \cdots, p_{N_a}]$ by the soft-max layer. The classification loss is formulated as

$$\mathcal{L}_a = -\sum\nolimits_{i=0}^{N_a} y_i \log p_i. \tag{16}$$

Taking all losses together, we have

$$\mathcal{L} = \mathcal{L}_c + \lambda_s \mathcal{L}_s + \lambda_a \mathcal{L}_a, \tag{17}$$

where $\lambda_s$ and $\lambda_a$ are constant parameters.

## 4    Experiments

We conduct experiments on the ActivityNet Captions[10] dataset that has been applied as the benchmark for dense video captioning. This dataset contains 20,000 videos in total, covering a wide range of complex human activities. For each video, the temporal segment and caption sentence of each human event is annotated. On average, there are 3.65 events annotated among each video, resulting in a total of 100,000 events. We follow the suggested protocol by [10, 11] to use 50% of the videos for training, 25% for validation, and 25% for testing.

The vocabulary size for all text sentence is set to be 6000. As detailed in the supplementary material, both the video and sentence encoders apply the GRU models[33] for feature extraction, where the dimensions of hidden and output layers are 512. The trade-off parameters in our loss, *i.e.*, $\lambda_s$ and $\lambda_a$ are both set to 0.1. We train our model by using the stochastic gradient descent with the initial learning rate as 0.01 and momentum factor as 0.9. Our code is implemented by Pytorch-0.3.

**Training.** Under the weak supervision constraint, the ground truth temporal segments are unused for training. The video itself is regarded as a special segment that is given by $S^{(f)} = (0.5, 1)$. We first pre-train the caption generator by using the entire video as input and each event caption among it as output. Such a pretraining process allows us to learn a well-initialized caption generator since the whole video content is related to the event caption, even the correlation is not precise. After the pretraining, we train our model in 2 stages. In the first stage, we minimize the captioning loss $\mathcal{L}_c$ and reconstruction loss $\mathcal{L}_s$. Then we minimize $\mathcal{L}_a$ in the second stage. Details about training are provided in the Supplementary materials and our Github repository.

**Testing.** For testing, only input videos are available. As already discussed in § 3.1, We starts from a random bunch of segments $\{S_i^{(r)}\}_{i=0}^{N_r}$ for initial guesses($N_r = 15$ in our reported result). After the one-round fixed-point iteration, we obtain the refined segments as $\{S_i\}_{i=0}^{N_r}$. We further filter them based on the IoU between $S_i^{(r)}$ and $S_i$(More details are given in the supplemental material), and keep those having high IoU as valid proposals. We then input the filtered segments to the caption generator to obtain event captioning sentences. It's nothing to mention that we do not choose using pretrained temporal segment proposal model(e.g. SST][22]) for the initial temporal segment generation, which, as a matter of fact, uses external temporal segment data, and is in contradiction with our motivation.

## 4.1 Evaluation of dense event captioning

**Evaluation metric.** The performance is measured with the commonly-used evaluation metrics: METEOR [32], CIDEr[23], Rouge-L[34], and Bleu@N[35]. We compute above metrics on the proposals if their overlapping with the ground-truth segments is larger than a given tIoU[3] threshold, and set the score to be 0 otherwise. All scores are averaged with tIoU thresholds of 0.3, 0.5, 0.7 and 0.9 in our experiments. We use the official scripts[4] for the score computation.

**Baselines.** Not any previous method is proposed for dense event captioning under the weak supervision. For Oracle comparisons, we still report the results by two fully-supervised methods [10, 11]. As for our method, we implement various variants to analysis the impact of each component. The first variant is the pretrained model where we randomly sample an event segment from each video and feed it into the pretrained caption generator for captioning in the testing phase. Another variant is the method by removing the anchor classification in Eq. 10, and thus regressing the temporal coordinate globally as in [29]. As a compliment, we also carry out the version by preserving the classification term but removing the regression component $\Delta S$ from Eq. 10.

**Results.** The event captioning results are summarized in Table 1. In general, the Meteor and Cider metrics are considered to be more convictive than other scores: the Meteor score is highly correlated with human judgment, and has been used as the final ranking in the ActivityNet challenge; while Cider is a newly proposed metric where the repetition of sentences is taken into account. Our method reaches comparable performance with the fully-supervised methods regarding the Meteor score and obtains the best score in terms of the Cider metric. Such results are encouraging as our method is weak supervised and not any

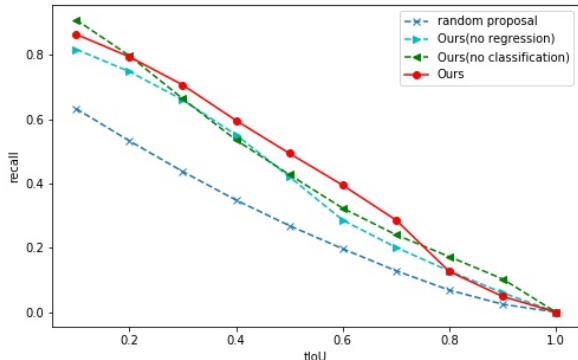

Figure 2: Evaluation of the event detection.

ground-truth segment is used. For the comparisons between different variants of our method, it is observed that removing the anchor classification or regression does decrease the accuracy, which verifies the necessity for each component in our model.

As we use a bunch of randomly selected temporal segments to generate the caption results, the robustness of the model towards such random strategy should also be evaluated. We use a different number of temporal segments and different random seeds to generate event caption sentences, and the evaluation results are summarized in Table 3. From the table, we can see that the variance is small on different random seeds. Besides, we can see a slight increase of performance along with the increase of the number of temporal segments. We choose $N_r = 15$ as a trade-off between complexity and performance in our final experiment.

Moreover, we display the recalls of the detected events by various methods with respect to the testing segments in Figure 2. To compute the recall, we assign the predicted segment as a positive sample if its overlap with the testing segment is larger than the tIOU threshold. From Fig. 2, we can find that our model is much better than the random proposal model, which verified the power of our weakly-supervised methods. Also, our final model is better than the two baselines in general.

**Illustrations.** Figure. 3 illustrates event captioning results of two videos. It presents the ground-truth descriptions, the captioning sentences by the pretrained model and our method. Compared with the pretrained model which generates a single description for each video, our model is capable to generate more accurate and detailed description. Compared to the ground truths, some of the descriptions are comparable in consideration of the generated sentence and event temporal segment. However, two issues still remain. One is that our model sometimes cannot capture the beginning of an event, which, in our opinion, is due to the fact that we use the final hidden state of a temporal segment to generate description which does not rely much on the starting coordinate. Another is that our model tris to

Table 1: Evaluation results of captioning. The term *ws* denotes "weak supervision" for short.

| Model | ws | M | C | R | B@1 | B@2 | B@3 | B@4 |
|---|---|---|---|---|---|---|---|---|
| Krishna's[10] | False | 4.82 | 17.29 | – | 17.95 | 7.69 | 3.86 | 2.20 |
| Yao's[11] | False | 7.71 | 16.08 | 13.27 | 17.50 | 9.62 | 5.54 | 3.38 |
| Pretrained | True | 4.58 | 10.45 | 9.27 | 8.7 | 3.39 | 1.50 | 0.69 |
| Ours (no classification) | True | 6.08 | 15.1 | 12.25 | 11.85 | 4.67 | 1.90 | 0.80 |
| Ours (no regression) | True | 6.11 | 17.66 | 12.40 | 11.98 | 5.45 | 2.69 | 1.44 |
| Ours | True | 6.30 | 18.77 | 12.55 | 12.41 | 5.50 | 2.62 | 1.27 |

Table 2: Evaluation results of sentence localization. The term *us* denotes "unsupervised" for short.

| Model | us | R@1, IoU 0.1 | R@1, IoU 0.3 | R@1, IoU 0.5 | mIoU |
|---|---|---|---|---|---|
| CTRL[28] | False | 49.09 | 28.70 | 14.00 | 20.54 |
| ABLR[29] | False | 73.30 | 55.67 | 36.79 | 36.99 |
| Full-supervised | False | 70.01 | 52.89 | 37.61 | 40.36 |
| Our Final | True | 62.71 | 41.98 | 23.34 | 28.23 |

generate 2 to 3 three descriptions most of the time, which means that it's not good at capture all the event in a video, especially those ones with many weeny events.

## 4.2 Evaluation of sentence localization

Using the learned caption localizer, our model can also be applied to the sentence localization task in an unsupervised way. In this section, we provide experimental results to demonstrate the effectiveness of our model on this task.

**Evaluation metric.** Following the works of [29, 28], we compute the "R@1, IoU=$\sigma$" and "mIoU" scores to measure the model's sentence localization ability. In details, for a given sentence and video pair, the "R@1, IoU=$\sigma$" score indicates the percentage of sentences who's top-1 predicted temporal segment has a higher IoU with the ground truth temporal segment than the given threshold $\sigma$, while the "mIoU" is the average tIoU between all top-1 prediction and ground truth temporal segment. In our experiment, $\sigma$ is set to 0.1, 0.3 and 0.5 following the setting in [29].

**Baselines.** We compare our model's sentence localization ability with Cross-modal Temporal Regression Localizer (CTRL) [28] and Attention Based Location Regression (ABLR) [29]. Such two models achieve the state-of-the-art performance for now. Besides the unsupervised model, we also implement a fully-supervised version by using ground-truth segments.

**Results** Table. 2 shows the results of all compared methods. First, our supervised implementation reaches similar performance as ABLR( the state-of-the-art) compared with another fully-supervised baseline, thus indicating the effectiveness of our model. As for the unsupervised scenario, we can see that our unsupervised model outperforms CTRL by a considerable margin, which shows that our model can really learn to locate meaningful temporal segment from the indirect losses.

Table 3: Evaluation of model rebustness towards random temporal segments during testing(see Sec. 4). We report the captioning evaluations on varying $N_r$. For each value of $N_r$, we run the experiments over 5 trials, and obtain the results in the form of *mean±std*.

| $N_r$ | M | C | B@1 | B@2 | B@3 | B@4 |
|---|---|---|---|---|---|---|
| $N_r = 10$ | 6.13±0.03 | 17.75±0.12 | 12.10±0.06 | 5.33±0.05 | 2.50±0.03 | 1.20±0.01 |
| $N_r = 15$ | 6.29±0.01 | 18.65±0.14 | 12.39±0.04 | 5.49±0.02 | 2.58±0.03 | 1.23±0.02 |
| $N_r = 20$ | 6.34±0.01 | 19.14±0.05 | 12.52±0.02 | 5.57±0.02 | 2.61±0.02 | 1.25±0.02 |

## 5 Conclusion and Future Work

We raise a new task termed Weakly Supervised Dense Event Caption(WS-DEC) and propose an efficient method to tackle it. The weak supervision is of great importance as it eliminates the source-consuming annotation of accurate temporal coordinates and encourages us to explore the huge amount of videos in the wild. The proposed solution not only solves the task efficiently but also provides an unsupervised method for sentence localization. Extensive experiments on both tasks verify the effectiveness of our model. For future research, one potential direction is to verify our model by performing experiments directly on Web videos. Meanwhile, since weakly supervised learning is becoming an important research vein in the domain, our proposed method by using the cycle process and fixed-point iteration could be applied to more other tasks, *e.g.*, weakly-supervised detection.

| Ground Truth | Pretrain model | Our final model |
| --- | --- | --- |

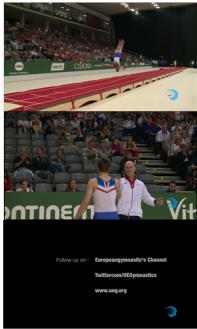

**Ground Truth**

A man begins running down a track and doing several flips and tricks while many watch on the sides.

He high fives another man and the same jumps are seen again in slow motion.

**Pretrain model**

a gymnast is seen running down a long track and jumping into the air

**Our final model**

she does several flips and tricks on the mat.

the video leads into several clips of people performing impressive jump roping with one another.

a black screen appears with white words on the screen.



**Ground Truth**

A man is standing on a field

The guy picks up the dog and carries it.

He throws frisbees and plays with a dog

**Pretrain model**

a man is seen standing in a field with a dog on his back and the dog jumps on his

**Our final model**

a man is seen standing in a large field with a dog and throwing a frisbee

the man throws the frisbee and the dog catches it

a man is seen standing in a large field with a dog and throwing a frisbee



**Ground Truth**

Several shots are shown of a man speaking to various groups of people and leads into people wearing wet suits and walking

The people walk down a beach and are seen swimming around in the water.

More shots of fish are shown and ends with the people walking out of the water and high fiving the camera man.

**Pretrain model**

a man is swimming in the water

**Our final model**

then , the man puts his helmet on the car and begins talking to the camera.

several shots of people are shown swimming in the ocean as well as a man swimming in the water.

Figure 3: Illustration of the generated dense event captions. Left is the ground truth, middle is the generation of our pretrained caption model, and right is by our weakly-supervised training approach. Different colors indicate different temporal segments and sentence descriptions.

**Acknowledgements**

This work was supported in part by National Program on Key Basic Research Project (No. 2015CB352300), and National Natural Science Foundation of China Major Project (No. U1611461).

## Footnotes

[3]temporal Intersection over Union

[4]https://github.com/ranjaykrishna/densevid_eval

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
