[Supplementary Material]

# Supplementary Material to:
# Weakly Supervised Dense Event Captioning in Videos

This Supplementary material provides proof of proposition 1, and presents more details about the model, training, and testing.

## A  Proof of proposition 1(Fixed-point iteration)

**Proposition 1** (Fixed-Point-Iteration). *We define the iteration as*

$$\boldsymbol{S}(t+1) \quad = \quad l_{\theta_1}(\boldsymbol{V}, g_{\theta_2}(\boldsymbol{V}, \boldsymbol{S}(t))), \tag{18}$$

*where $\boldsymbol{S}(t)$ will converge to the fixed-point solution,* i.e. $\boldsymbol{S}^* = l_{\theta_1}(\boldsymbol{V}, g_{\theta_2}(\boldsymbol{V}, \boldsymbol{S}^*))$, *if the initial start $\boldsymbol{S}(0)$ surrounds the fixed point $\boldsymbol{S}^*$ sufficiently and the function $l_{\theta_1}(\boldsymbol{V}, g_{\theta_2}(\boldsymbol{V}, \boldsymbol{S})$ is locally Lipschitz continuous around $\boldsymbol{S}^*$ with Lipschitz constant $L < 1$.*

*Proof.* Since the function $f(\boldsymbol{S}) = l_{\theta_1}(\boldsymbol{V}, g_{\theta_2}(\boldsymbol{V}, \boldsymbol{S})$ is Lipschitz continuous with Lipschitz constant $L \le 1$, we have,

$$
\begin{aligned}
|\boldsymbol{S}(t+1) - \boldsymbol{S}(t)| &= |f(\boldsymbol{S}(t)) - f(\boldsymbol{S}(t-1))| \\
&\le L|\boldsymbol{S}(t) - \boldsymbol{S}(t-1)|, \\
&\vdots \\
&\le L^t|\boldsymbol{S}(1) - \boldsymbol{S}(0)|.
\end{aligned}
\tag{19}
$$

Since $L < 1$, $|\boldsymbol{S}(t+1) - \boldsymbol{S}(t)| \to 0$ converges to zeros as $t \to \infty$. Thus, $\boldsymbol{S}(t)$ will converge to the fixed point that is the solution of Eq.(2) in the paper. □

## B  Video Encoder & Sentence Encoder

In § 3, we have introduced the details of our model except the RNN-based video and sentence encoder used in feature extracting, which are considered not crucial in the paper. But for reproduction of our work, we will introduce them detailedly in this section.

### B.1  Video Encoder.

Reviewing § 3, the video encoder aims to encode the video frames into a set of vectors. For this purpose, a C3D[1] network is constructed to extract frame-block features and resolve short-term dependency, while a followed GRU[2] is leveraged for long-term dependency.

We denote the input video as $\boldsymbol{V} = \{\boldsymbol{f}_t\}_{t=0}^{T_v}$ where $T_v$ is the temporal length of the video and $\boldsymbol{f}_t$ specifies the $t$-th frame. Following the implementation by[3], we first extract the non-overlapping C3D features from the original image frames with a interval $\delta$, i.e., computing $\{\boldsymbol{x}_t^{(v)} = C3D(\boldsymbol{f}_{(t-1)*\delta+1} : \boldsymbol{f}_{t*\delta})\}_{t=0}^{T_v/\delta}$, where $\delta = 16$.

The extracted C3D features are fed into a GRU network to learn the long-term dependence. Specially, we set the initial hidden state as zero, i.e. $\boldsymbol{h}_0^{(v)} = \boldsymbol{0}$, and compute the following outputs and hidden states recursively by

$$\boldsymbol{v}_t, \boldsymbol{h}_t^{(v)} \quad = \quad GRU_{video}(\boldsymbol{x}_t^{(v)}, \boldsymbol{h}_{t-1}^{(v)}). \tag{20}$$

The output sequence $\{\boldsymbol{v}_t\}_{t=0}^{T_v/\delta}$ and hidden states $\{\boldsymbol{h}_t^{(v)}\}_{t=0}^{T_v/\delta}$ are used in the followed processes of sentence localizing and caption generating as video representation (In the original paper, we omit $\delta$ and denote $T_v/\delta$ as $T_v$ and the extract feature as $\{\boldsymbol{v}_t\}_{t=0}^{T_v}$ for simplify).

## B.2 Sentence Encoder

Similar to the video encoder, sentence encoder aims to encode natural sentences into vector representations. Given a sentence $\boldsymbol{C} = \{\boldsymbol{w}_t\}_{t=0}^{T_c}$, $\boldsymbol{w}_t \in \{0,1\}^V$ denote the one-hot encoding of the $t$-th word, where $V$ is the vocabulary size. Then, $\boldsymbol{w}_t$ is embedded into a $k$-dimensional vector by $\boldsymbol{x}_t^{(c)} = \boldsymbol{W}_e \boldsymbol{w}_t$, with $\boldsymbol{W}_e \in \mathcal{R}^{k \times V}$ being the trainable embedding matrix. Similar to the video encoder, the embedded features $\{\boldsymbol{x}_t^{(c)}\}_{t=0}^{T_c}$ are fed into a GRU model. Specially, the initial hidden state $\boldsymbol{h}_0^{(c)}$ is set to be zero, and the outputs and following hidden states are computed as:

$$\boldsymbol{c}_t, \boldsymbol{h}_t^{(c)} = GRU_{text}(\boldsymbol{x}_t^{(c)}, \boldsymbol{h}_{t-1}^{(c)}). \tag{21}$$

After the encoding stage, the final outputs $\{\boldsymbol{c}_t\}_{t=0}^{T_c}$ and hidden states $\{\boldsymbol{h}_t^{(c)}\}_{t=0}^{T_c}$ are used in the followed networks.

## B.3 More Details

**I.** As both **Sentence Localizer** $l_{\theta_1}$ and **Caption Generator** $g_{\theta_2}$ use the video features for further processing, they can choose whether to share the same video encoder parameters or not. In our previous experiments, we found that both strategies reach similar performances. The reported results are obtained using different video encoder parameters.

**II.** Also, as training a C3D network from scratch is very time-consuming and error-prone, we directly adopt the public-available C3D features[1]. The public-available features are 500-way features reduced from the original 4096-way output of C3D's fc7 layer using PCA. In our experiments, the features are denoted as $\{\boldsymbol{x}_t^{(v)}\}_{t=0}^{T_v/\delta}$ and fed into the $GRU_{video}$ for further processing.

# C  Training & Testing Details

## C.1  Training

Because our model consists of two submodels and several term losses, there may be several ways to train the whole model. In this section, we introduce the strategy we used in our experiments. Firstly, the model is pretrained with our predefined *Fake Proposal*, i.e. $\boldsymbol{S}^{(f)} = (0.5, 1)$. After pretraining for several rounds, we train the proposed three term losses alternatively. Details are shown in Algorithm1.

## C.2  Testing

As our model is not directly trained on the dense event captioning problem, we provide an extra explanation on the strategy we used in testing. In short, our training strategy and losses force the model meets the requirement of fixed-point iteration, so we use a random bunch of segments $\{\boldsymbol{S}_i^{(r)}\}_i^{N_r}$ as initial temporal segments, feed them into the cycle system $\boldsymbol{S}_i' = l_{\theta_1}(\boldsymbol{V}, g_{\theta_2}(\boldsymbol{V}, \boldsymbol{S}_i^{(r)}))$. Considering that those random segments will converge to the true event temporal segments or diverge to some unknown random segment, the distance between $\boldsymbol{S}_i'$ and $\boldsymbol{S}_i^{(r)}$ should be small if they are in the neighborhood of a certain event. Also, we measure the distance between $\boldsymbol{S}_i'$ and $\boldsymbol{S}_j'$ to further reduce redundancy(This step is not critical, but can reduce the repetition of temporal segments.). Specifically:

- if $dist(\boldsymbol{S}_i', \boldsymbol{S}_i^{(r)}) > \Theta_1$, remove $\boldsymbol{S}_i'$ from the predicted temporal segment set

- if $dist(\boldsymbol{S}'_i, \boldsymbol{S}'_j) < \Theta_2$, merge $\boldsymbol{S}'_i, \boldsymbol{S}'_j$ as: $\boldsymbol{S}''_i = union(\boldsymbol{S}'_i, \boldsymbol{S}'_j)$

where $dist(\cdot)$ computes the IoU between $\boldsymbol{S}_1, \boldsymbol{S}_2$:

$$dist(\boldsymbol{S}_1, \boldsymbol{S}_2) = \frac{intersection(\boldsymbol{S}_1, \boldsymbol{S}_2)}{union(\boldsymbol{S}_1, \boldsymbol{S}_2)} \tag{22}$$

Details about testing are shown in Algorithm2.

---

**Algorithm 1** Training pipeline for the WS-DEC model

---

**Input:** $\mathcal{D}$      // the dataset iterator for training
**Input:** $l_{\theta_1}, g_{\theta_2}$      // the random initialized sentence localizer, caption generator
**Input:** $\{\boldsymbol{S}^{(a)}_j\}^{N_a}_{j=0}$      // anchor segments for training $l_{\theta_1}$
**Input:** $\boldsymbol{S}^{(f)}$      // $\boldsymbol{S}^{(f)} = (m^{(f)}, w^{(f)})$ is used to pretrain the caption generator
**Output:** $\theta_1, \theta_2$      // trained parameters for sentence localizer and caption generator

1: $step \leftarrow 0$
2: **while** $step < pretrain\_step$ **do** // we pretrain the model with fake proposal
3:      **for** $(\boldsymbol{V}, \{\boldsymbol{C}i\}^{N_v}_{i=0}) \in \mathcal{D}$ **do**
4:          $\boldsymbol{C} \leftarrow$ RANDOMCHOOSE$(\{\boldsymbol{C}i\}^{N_v}_{i=0})$ // randomly choose a sentence
5:          $\boldsymbol{C}' \leftarrow g_{\theta_2}(\boldsymbol{V}, \boldsymbol{S}^{(f)})$ // obtain the fake generation
6:          $\mathcal{L}_c \leftarrow dist(\boldsymbol{C}, \boldsymbol{C}')$ //compute the loss
7:          $\theta_2 \leftarrow \theta_2 - \eta_2 \frac{\partial \mathcal{L}_c}{\partial \theta_2}$ // update parameters with SGD
8:      **end for**
9:      $step \leftarrow step + 1$
10: **end while**
11: **while** $step < training\_step$ **do** // we train the model with three term losses alternatively
12:      **if** training $L_c$ and $L_s$ **then** // train the model with $\mathcal{L}_c$ and $\mathcal{L}_s$
13:          **for** $(\boldsymbol{V}, \{\boldsymbol{C}i\}^{N_v}_{i=0}) \in \mathcal{D}$ **do**
14:              $\boldsymbol{C} \leftarrow$ RANDOMCHOOSE$(\{\boldsymbol{C}i\}^{N_v}_{i=0})$
15:              $\boldsymbol{S}' \leftarrow l_{\theta_1}(\boldsymbol{V}, \boldsymbol{C})$
16:              $\boldsymbol{C}' \leftarrow g_{\theta_2}(\boldsymbol{V}, \boldsymbol{S}' + \delta)$
17:              $\boldsymbol{S}'' \leftarrow g_{\theta_2}(\boldsymbol{V}, \boldsymbol{C}')$
18:              $\mathcal{L} \leftarrow dist(\boldsymbol{C}, \boldsymbol{C}') + dist(\boldsymbol{S}', \boldsymbol{S}'')$
19:              $\theta_1 \leftarrow \theta_1 - \eta_{c1} \frac{\partial \mathcal{L}}{\partial \theta_1}$
20:              $\theta_2 \leftarrow \theta_2 - \eta_{c2} \frac{\partial \mathcal{L}}{\partial \theta_2}$
21:          **end for**
22:      **end if**
23:      **if** training $L_a$ **then** // train the model with $\mathcal{L}_a$
24:          **for** $(\boldsymbol{V}, \{\boldsymbol{C}i\}^{N_v}_{i=0}) \in \mathcal{D}$ **do**
25:              $\boldsymbol{C} \leftarrow$ RANDOMCHOOSE$(\{\boldsymbol{C}i\}^{N_v}_{i=0})$
26:              $\{\boldsymbol{C}j^{(a)}\}^{N_a}_{j=0} \leftarrow \{g_{\theta_2}(\boldsymbol{V}, \boldsymbol{S}^{(a)}_j)\}^{N_a}_{j=0}$ //anchor segments $\rightarrow$ anchor sentences
27:              $confidence \leftarrow l_{\theta_1}(\boldsymbol{V}, \boldsymbol{C})$ // confidence about each anchor regarding $\boldsymbol{V}$ and $\boldsymbol{C}$
28:              $\{score^{(a)}_j\}^{N_a}_{j=0} \leftarrow \{$ METEOR$(\boldsymbol{C}, \boldsymbol{C}j^{(a)}) \}^{N_a}_{j=0}$ // compute Meteor score
29:              $label \leftarrow \arg\max_j \{score^{(a)}_j\}^{N_a}_{j=0}$ // the one with highest score is considered as label.
30:              $\mathcal{L}_a \leftarrow$ CROSSENTROPY$(confidence, label)$
31:              $\theta_1 \leftarrow \theta_1 - \eta_{a1} \frac{\partial \mathcal{L}_a}{\partial \theta_1}$ // cross entropy loss
32:          **end for**
33:      **end if**
34:      $step \leftarrow step + 1$
35: **end while**
36: **return** $\theta_1, \theta_2$

---

**Algorithm 2** Testing pipeline for the WS-DEC model

---

**Input:** $V$    // a video for testing(never seen in the training set)
**Input:** $l_{\theta_1}, g_{\theta_2}$    // the learned sentence localizer, caption generator
**Input:** $\{S_i^{(r)}\}_{i=0}^{N_r}$    // initial random bunch temporal segments(fixed for all testing example)
**Output:** $\{C_i, S_i\}_{i=0}^{N_e^{(v)}}$    // the predicted events within the video
1: $step \leftarrow 0$ // iteration round
2: $N_e^{(v)} \leftarrow N_r$ // initial number of event equals the predefined one
3: $\{S_i\}_{i=0}^{N_e^{(v)}} \leftarrow \{S_i^{(r)}\}_{i=0}^{N_r}$ // initial temporal segment equals the predefined temporal segments
4: $\{C_i\}_{i=0}^{N_e^{(v)}} \leftarrow \{g_{\theta_2}(V, S_i)\}_{i=0}^{N_e^{(v)}}$ // generate initial description
5: **while** $step < iteration\_step$ **do** // iterate until max iteration step
6:     $\{S_i'\}_{i=0}^{N_e^{(v)}} \leftarrow \{l_{\theta_1}(V, C_i)\}_{i=0}^{N_e^{(v)}}$ // re localize the generated sentence
7:     $\{S_i''\}_{i=0}^{N_e^{(v)'}} \leftarrow \text{SEGMENTMERGE}(\{S_i\}_{i=0}^{N_e^{(v)}}, \{S_i'\}_{i=0}^{N_e^{(v)}})$ // merge segments based on§ 4.3.3
8:     $N_e^{(v)} \leftarrow N_e^{(v)'}$ // update the predicted number of events
9:     $\{S_i\}_{i=0}^{N_e^{(v)}} \leftarrow \{S_i''\}_{i=0}^{N_e^{(v)}}$ // update events temporal segments
10:    $\{C_i\}_{i=0}^{N_e^{(v)}} \leftarrow \{g_{\theta_2}(V, S_i)\}_{i=0}^{N_e^{(v)}}$ // update description sentences
11:    $step \leftarrow step + 1$
12: **end while**
13: **return** $\{C_i, S_i\}_{i=0}^{N_e^{(v)}}$

---

## Footnotes

[1]http://activity-net.org/challenges/2016/download.html#c3d