[Reviews · NeurIPS 2018]

Reviewer 1



This paper tackles the task of event captioning in videos. Traditional event captioning methods follow “localization and captioning” schema, while this paper proposes to nest them together and optimized in an iterative fashion. While the former needs segment-wise caption annotation, the later only needs video-wise sentences annotation, thus, a weakly supervised learning is achieved. Based on proposed a novel fixed-point iteration algorithm, the experimental results on ActivityNet Captions dataset achieves somewhat comparable results as supervised alternatives. This paper is clearly written, with minor typos exist. In general, it is easy to follow. Strengths: The idea solving dense event captioning as a weakly supervise problem by intertwist sentence localization and captioning is novel. This potentially helps to reduce annotation cost. Weaknesses: 1) The implementation of l_θ1 and g_θ2 are stacked by multiple neural layers, which not naturally satisfy the local-Lipschitz-continuity in Proposition 1 (As it mentioned in line 136-137). The author proposed to smooth the mapping by add Gaussian noise to training data. But I don’t see how this trick can guarantee the satisfaction. Thus, there’s risk of non-converging iteration. 2) Why l_θ1 and g_θ2 satisfy Lipschitz constant L<1? The sentence “if the initial start S(0) surrounds … sufficiently” in Proposition 1 is a little bit vague. 3) The proposed method stands on an important assumption that there’s a one-to-one correspondence of caption sentence and temporal segment in video. This mainly relates to what data is used during experiments. Thus, it would be worthwhile to show some evidence why this assumption is valid. 4) During test time, a random initial point is needed.

Reviewer 2



This paper addresses the problem of weakly-supervised dense event captioning (WS-DEC), where there are only video captioning labels without temporal segmentation labels. Utilizing the one-to-one correspondence assumption, it proposes a new iterative formulation for WS-DEC that alternates between caption generation and sentence localization. The proposed algorithm is evaluated on ActivityNet Captions benchmark. + This paper tackles the weakly-supervised dense event captioning (WS-DEC), which extends the existing DEC problem by excluding the requirement of temporal segmentation labels. This WS-DEC task has not been discussed in previous research. + The problem formulation seems well-defined and effective (section 3.1). The proposed approach is designed by alternating two subproblems, sentence localization and event captioning, both of which are formulated to be mathematically clear. + The paper reads very well. 1. I am not sure the term WS-DEC is correct for the problem of this paper. - While the DEC formulation requires two types of labels for sentences and temporal localization, the proposed WS-DEC exploits only sentence labels. Obviously, sentence labels are much more informative than localization labels (only two numbers ranged in [0,1]). 2. The proposition 1 (Fixed-point-iteration) is dubiously and invalidly applied to the WS-DEC formulation. - First of all, citation is required for the proposition 1. I don’t think the convergence proof of fixed-point algorithm with Lipschitz continuity is on authors’ contribution. I cannot any authors’ novelty here. It is a simple re-utterance of the existing proposition with only new notations for the functions in Eq.(1)-(2). - The functions that this paper uses are not Lipschitz continuous (Line 137); thus, this proposition is not applicable to the problem formulation of this paper. - Moreover, this paper argues that only a single-round iteration is enough to get a reasonable solution empirically (Line 140), which may hint that the fixed point-iteration is neither necessary nor sufficient for the problem. - In summary of my opinion, this proposition could be mentioned as a background rationale for the proposed iterative algorithm, but strictly speaking it does not seem to logically or mathematically connected to the formulation of WS-DEC. 3. Some analyses on experimental evaluation seems hasty and defective. - As shown in Table 1 and 2, the proposed unsupervised algorithm works better than some initial supervised models (e.g. [10, 28]) but underperforms the best-performing methods (e.g. [11.29]). This tendency should be explicitly mentioned in the paper. - I do not think the statement in Line 276-277 is correct. The supervised implementation works worsen than ABLR with nontrivial margins. 4. Minor comments - How can we set the N_r (Line 130), which seems critical for the final output? - What is the rationale behind choosing perplexity loss for text reconstruction? (Line 185) It is for performance or implementation easiness?

Reviewer 3



This paper presented the new task of event captioning in videos with weak supervision, where the set of captions is available but not the temporal segment of each event. To tackle this problem, the authors decompose the global problem into a cycle of dual problems: event captioning and sentence localization. This cycle process repeatedly optimizes the sentence localizer and the caption generator by reconstructing the caption. The performances are evaluated on the ActivityNet Captions dataset. I have several concerns about this paper: - The motivation of this task is not clear. For which applications is it easy to get the event captions without the temporal segment? - I am not convinced by the testing procedure where random segments are used to generate a set of event captions for each video because a temporal segment can overlap several events or no event. I think that using pre-trained temporal segment proposal like [101] could be better. - The authors should analyze the sensibility to the initial candidate segments because they are generated randomly? The authors should report standard deviation in the results. - The authors should give information about the hyperparameters tuning and how the model is sensible to these hyperparameters. - The authors should explain how they chose the number of candidate temporal segments in test videos. This number seems important because they used one-to-one correspondence strategy. - The authors proposed Crossing Attention mechanism but they did not analyze the importance of this attention mechanism. The authors should compare Crossing Attention mechanism with existing attention mechanisms. The proposed method combines several existing works, but the results seems interesting. However this new task is not well motivated and I am not convinced by the testing procedure where random candidate temporal segments are generated. [101] Shyamal Buch and Victor Escorcia and Chuanqi Shen and Bernard Ghanem and Juan Carlos Niebles. SST: Single-Stream Temporal Action Proposals. In CVPR, 2017. I have read the author response and I am globally satisfied. But I am still not completely convinced that generating random temporal segments is a good strategy. However the authors show that their model is robust to these random segments. I think that using temporal segment proposal trained on external data can be useful to generate better segment.